# Surface Quality and Material Removal Rate in Fabricating Microtexture on Tungsten Carbide via Femtosecond Laser

**DOI:** 10.3390/mi14061143

**Published:** 2023-05-28

**Authors:** Guangxian Li, Xuanang Li, Guichao He, Ruiguang Fan, Feiyuan Li, Songlin Ding

**Affiliations:** 1School of Mechanical Engineering, Guangxi University, Nanning 530004, China; liguangxian@gxu.edu.cn (G.L.); 2001300611@st.gxu.edu.cn (F.L.); 2School of Engineering, RMIT University, Victoria 3083, Australia

**Keywords:** tungsten carbide, femtosecond laser, microtexture, material removal rate, laser-induced periodic surface structure

## Abstract

Tungsten carbide is currently the most widely used tool material for machining difficult-to-machine materials, such as titanium alloys and nickel-based super alloys. In order to improve the performance of tungsten carbide tools, surface microtexturing, a novel technology that can effectively reduce cutting forces and cutting temperatures and improve wear resistance, has been applied in metalworking processes. However, when fabricating the micro-textures such as micro-grooves or micro-holes on tool surfaces, the significant decrease in material removal rate is a major obstacle. In this study, a straight-groove-array microtexture was fabricated on the surface of tungsten carbide tools via a femtosecond laser with different machining parameters including laser power, laser frequency, and scanning speed. The material removal rate, surface roughness, and the laser-induced periodic surface structure were analyzed. It was found that the increase in the scanning speed decreased the material removal rate, whereas increasing the laser power and laser frequency had the opposite effects on the material removal rate. The laser-induced periodic surface structure was found to have a significant influence on the material removal rate, and the destruction of the laser-induced periodic surface structure was the reason for the reduction in the material removal rate. The results of the study revealed the fundamental mechanisms of the efficient machining method for the fabrication of microtextures on ultrahard materials with an ultrashort laser.

## 1. Introduction

Tungsten carbide (WC) tools are the most widely used cutters in different metal-cutting processes [1,2] due to their exceptional mechanical properties such as high hardness, good abrasive resistance, and heat endurance [3,4,5]. However, with the advance of material engineering and additive manufacturing technology, many new alloys are being developed, and the high temperatures [6] and dynamic loads [7,8,9] in cutting these advanced alloys lead to severe adhesive–abrasive wear on the tool surface [10,11]. Therefore, new surface treatments of tungsten carbide tools are required to enhance their performance. Inspired by biomimetics, some bionic micro- or nanostructures imitating the patterns of animal skin or plant surfaces have recently been developed [12,13]. These micro/nanostructures can change the frictional characteristics at the tool/chip interface and tool/workpiece contact regions, reducing the cutting force and tool/chip/workpiece abrasion and adhesion, eventually improving the wear resistance of the cutting tools [14,15]. The bionic structures of the tool surface are generally at the micro or nano level. Because WC is a kind of brittle material [16] with relatively high hardness, high-quality and precise bionic structures can hardly be achieved via conventional machining processes, e.g., micromilling or microdrilling.

Nonconventional machining methods such as electrical discharge machining (EDM) [17,18], electrical discharge grinding [19,20], electrochemical machining (ECM) [21,22], and laser machining are effective methods in the machining of ultrahard materials. Compared with electrical machining methods, laser machining is often used in the fabrication of microstructures on the tool surface due to its unique advantages of being high precision and ecofriendly and having the capability to generate complex textures on free-form surfaces. There are plenty of studies about the laser-processed surface textures on different cutting tools. These studies have focused on the patterns of the micro-textures on tool surfaces and their influence on the performance of cutting tools. For example, Khani et al. [23] fabricated microholes and microgrooves on the rake face of tungsten carbide tools using a fiber laser. The performance of the textured tools was compared with that of conventional tools in the turning of aluminum alloy 7075. Their results showed that microholes led to better performance; and the optimum width, depth, and distance during thread turning process were 93.43 µm, 15 µm, 50 µm, respectively. Kawasegi et al. [24] created microgroove textures with various directions at the nano/micro scale on the rake faces of tungsten carbide tools via laser texturing. It was found that the cutting forces decreased when the direction of the texture was perpendicular to the cutting edge, whereas the cutting forces were slightly higher than those of conventional tools when the texture direction was parallel to the cutting edge. Similarly, Zheng et al. [25] produced laser-processed microgrooves on the surface of tungsten carbide tools and studied their tribological mechanisms through wear experiments. It was found that metal debris was promptly removed along with the movement of the friction pair when the microgroove direction was consistent with the friction direction. This indicated that it facilitated the ejection of debris and could reduce the adhesion of the workpiece material on the tool surface with this texture. However, when the microgroove direction was perpendicular to the friction direction, the performance of the microgrooves was minimized.

The optimization of laser parameters is a common research topic in improving the quality of microstructures. Yang et al. [26] conducted wear experiments on laser-textured surfaces and investigated the effects of the laser parameters on the surface wear rate at different friction distances. The study found that laser power was the main factor affecting the surface wear rate of a textured WC surface. In combination with the influences of the laser parameters, it was concluded that the laser power and the diameter of the microtexture had the greatest influence on the wear resistance, and the influence gradually decreased with the increase in the friction distance, whereas the number of scans and micro-texture diameter became more significant. However, the discussion on the relationship between the machining efficiency and the quality of the texture has been limited. Fatima et al. [27] studied the influence of the laser flux and scanning speed on the geometry and quality of textured structures. The results showed that the width and depth of the microtextures increased with the increase in the laser flux, whereas it decreased with the increase in the scanning speed. When the laser flux increased from 52 J/cm^2^ to 65 J/cm^2^, a slight decrease in the surface roughness was observed. In contrast, the surface roughness decreased when the scanning speed decreased from 20 mm/s to 10 mm/s, whereas the surface roughness increased again when the speed further decreased to 6 mm/s, 4 mm/s, 1 mm/s, and 0.5 mm/s. The authors analyzed the machining depth, width, roughness, hardness, material removal rate, and heat-affected zone in detail. However, they did not present the dynamic changes in the machining response with ongoing laser machining. Vazquez et al. [28] used a nanosecond laser to fabricate microgrooves on WC-Co at three laser speeds. The fastest laser speed (150 mm/s) only penetrated the topmost layer but still delivered enough energy to cause material spatter along the texture. This indicated that with the increase in laser speed, the intensity of the laser treatment was lost, providing softer topographies with shallower textured grooves. In addition, Vazquez et al. [29] studied the influence of the energy density of the pulse and the scanning speed on the track dimension, roughness, microstructure, hardness, and lubricant retention ability of modified surfaces. It was found that when the laser energy density was 30 J/mm^2^ and the scanning speed was 50 mm/s, the overlapping ratio between the areas of two consecutive laser pulses was 95%, and the groove depth more significantly increased than the width because of the insufficient cooling of the heated workpiece material. Both studies investigated the influence of laser parameters on the width and depth of microgrooves. However, no analysis of the material removal rate (MRR) was conducted, which implies that the optimal parameters could have limited effects on the improvement in machining efficiency. Chu et al. [30] investigated the effects of laser parameters on the evolution of surface morphology and the mechanism of surface damage. The results demonstrated that a negative correlation existed between the laser scanning speed (increased from 100 mm/s to 400 mm/s) and the increase in groove depth. Although increasing the scanning speed resulted in a reduction in the machining depth, the corresponding changes in the MRR were not necessarily consistent with this trend. Therefore, a comprehensive analysis of the temporal scale is needed to further investigate this phenomenon. Zhang et al. [31] utilized an ultraviolet nanosecond laser to fabricate radial arrays with widths of 40 µm, 70 µm, 100 µm, and 130 µm on a mirror-polished tungsten carbide disk. The investigation revealed when the laser power exceeded 7 W and the laser frequency was 1.5 MHz, due to the obstruction of the plasma, the laser energy progressively decreased at the bottom of the groove, resulting in a characteristic morphology with a wider upper region and a narrower lower region of the texture. Additionally, the difference in the melting point of WC and Co in the cemented carbide resulted in an uneven bottom. Similar to the previous studies, an investigation into the material removal rate was neglected. Liu et al. [32] conducted laser surface texturing (LST) experiments on tungsten carbide with different laser scanning speeds, frequencies, and powers. The results revealed that within the parameters ranges studied, the laser energy absorption coefficient for cemented carbides decreased with the increase in the laser frequency and increased with the increase in the laser power. Scanning speed had almost no effect on the absorption coefficient. However, the investigation did not cover high laser power or frequency. Excessively high laser power resulted in a significant heat-affected zone (HAZ) and induced plasma shielding effects, which significantly reduced the laser absorption rates. Lu et al. [33] fabricated linear microtextures on the surface of a hard alloy with a Nd:YAG laser with different incident angles. It was found that when the angle decreased from 90° to 30°, the depth of the microstructure, laser processing temperature, and the residual stress were reduced; and the width of the microstructure and the surface roughness were increased. As the incident angle caused changes in the laser energy density on the material surface, the observed variations in the microtexture patterns could be attributed to the reduction of energy density. This analysis offered valuable insight from another perspective on laser parameter issues. However, because the MRRs were not analyzed in the study, it was difficult to determine the impact of the changes in the incident angle on the processing efficiency.

From the results of the above research, it is clear that most of the studies have provided detailed analyses of the types and sizes of microtextures, the actual cutting performance, and the effects of laser processing parameters. Nevertheless, investigations into the evolution of the microstructure profile, surface roughness, and material removal rate have been limited. On one hand, the material removal rate dynamically changes with ongoing machining, and it could be the key factor that causes the variance in the final profile of a microstructure. On the other hand, the profile of the microstructure and the roughness of the machined surface can dynamically affect the material removal rate. Therefore, it is necessary to explore the changes in such machining responses with ongoing laser machining. In this study, a femtosecond pulse laser was applied to fabricate groove arrays with different parameters on a WC-Co surface. The changes in roughness, geometry, and material removal rate with ongoing laser machining for different laser parameters were investigated. The laser-induced periodic surface texture was investigated and found to play the key role in affecting the material removal rate. This approach provides a new perspective for enhancing the machining efficiency of tool microtexturing.

## 2. Experiments on Femtosecond Laser Ablation

The tool material used in this experiment was commercial tungsten carbide M20; its mechanical properties and chemical composition are listed in Table 1 and Table 2, respectively. Rectangular M20 specimens (3 mm × 10 mm × 20 mm) were shaped via wire-cut EDM. The surfaces of the workpiece were polished to a surface roughness of 1.6~3.2 µm to remove the oxidation layer. The samples were then processed via ultrasonic cleansing with an anhydrous ethanol bath and thoroughly dried. Finally, the specimens were preserved in a furnace to prevent further oxidation and contamination. The morphology and profile of the workpiece are shown in Figure 1.

A multiline straight groove array with the dimension of 3 mm × 4 mm was fabricated on the surface of the M20 samples using a femtosecond laser with different laser parameters. The experimental setup is presented in Figure 2. The femtosecond laser used in this study was a FemtoYL-20 laser (YSL Photonics Company Ltd, Wuhan, China), and the specifications of the laser are listed in Table 3. In the ablation experiments, eight tests were conducted with different laser parameters, as listed in Table 4, in order to investigate the effects of the laser parameters on the MRR and surface quality. The laser frequencies and scanning speed were the optimal values obtained in our preliminary experiment and those published in the literature [27,29]. The average power of the laser was determined with a fixed laser power density, and the values were measured with a Gentec-ED power meter (UP55N-300F-H12-INT). 

The depth of the groove after the processing cycles listed in Table 4 was measured to evaluate the material removal rate with ongoing ablation. The measurement was conducted using a super depth-of-field microscope (KEYENCE VHX-7000). The microstructures and grain structures inside the laser-ablated grooves were observed and analyzed using a scanning electron microscope (HITACHI UHR FE-SEM SU8000 Series). Because the minimum size measured in this study was 0.01 µm, which is within the nano scale, the presence of noise [34,35] and uncertainty [36] needed to be considered to ensure the reliability of the data. Therefore, with regard to the super depth-of-field microscope, we increased the image sampling frequency and optimized the obtained images using the inherent Accurate D.F.D. 2.0 and Auto Adjust function in the system, ultimately obtaining clear and accurate images. Regarding the scanning electron microscope, we performed observations at an acceleration voltage of 20 KV and in high-magnification amplification mode in order to obtain stable and clear results.

## 3. Experiment Results and Discussion

### 3.1. Surface Topography and Material Removal Rate

The changes in the surface roughness after different numbers of ablation cycles and the texture morphology of the microgrooves obtained with different laser parameters after 20 cycles are shown in Figure 3. Considering the laser processing parameters and the evolution of surface morphology, the changes could be classified into three categories: 

The first category was represented by Tests 2, 4–7 (Figure 3b,d–g), in which the Ras were 0.60 µm, 0.62 µm, 0.42 µm, 0.40 µm, and 0.42 µm after 20 processing cycles. It could be observed that, in these tests, the Ras generally initially presented decreasing trends and stabilized after 10 ablations. The values were relatively lower compared with those in the rest of the tests. The reason for this finding is that during processing with such parameters, the laser energy was not sufficient to produce microgroove textures with good surface quality after the first cycle of machining. As the processing cycles gradually increased to ≥10, the laser energy stabilized, allowing for continuous processing and maintaining a lower surface roughness level of the microgrooves.

The second category was represented by Sample 1 (Figure 3a), which had a surface roughness Ra of 1.57 µm after 20 processing cycles. It can be seen that its evolution trend was similar to that of the first category at lower numbers of processing cycles. However, the surface roughness drastically increased after 10 processing cycles. This could be ascribed to the weak laser power. Due to the different melting points of WC and Co in cemented carbide, different energies are required for their removal. As a result, uneven ablation may occur at lower laser energy and deeper processing depths, leading to uneven microgroove bottoms during the machining, which can subsequently result in an increasing trend in surface roughness with increasing numbers of machining cycles.

The third category was represented by Samples 3 and 8 (Figure 3c,h), which had a surface roughness Ra of 1.17 and 4.28 µm, respectively. Due to the low scanning speed and high laser frequency used in the machining, the same position received more laser energy than the other two categories. Therefore, after a single processing cycle, microgrooves with lower roughness could be obtained that stayed stable for a certain number of cycles before experiencing a sudden increase in roughness. The reason for this is that despite the small heat-affected zone and the limited heat accumulation, continuous processing under higher laser energy over longer periods of time over an increased number of processing cycles cause heat to accumulate and prevent efficient dissipation, resulting in excessive material ablation and significant damage to the surface topography, ultimately leading to a sharp rise in surface roughness.

Based on the evolution trends of the three types of roughness mentioned above, it is clear that the change in the inner roughness of the microgroove textures produced under different laser parameters followed a regular pattern. When the laser power and frequency were too high (11.20 W and 0.5 MHz, respectively) or too low (1.92 W and 0.2 MHz, respectively), the machining quality deteriorated after multiple cycles. However, in the processing with parameters in between these limits, the roughness tended to be better with the increase in the number of processing cycles. Notably, the roughness evolution of Sample 8 (Figure 3h) showed that its surface roughness was relatively low when the cycle number N was ≤8, and the machining effect at this point met the requirements. Therefore, analyzing the influence of laser parameters on roughness alone is insufficient, and further analysis is necessary to ensure high material removal rates without sacrificing surface quality.

The material removal rate is defined as the volume of tungsten carbide removed by a laser per unit time, which can be calculated using the following equation [37]:(1)$$MRR=\frac{V}{T}=\frac{Sh}{T}$$
where *MRR* is material removal rate (mm^3^/s); *V* is the volume of tungsten carbide removed by the laser (mm^3^); *S* is the area of tungsten carbide removed by laser (mm^2^); *h* is the depth of tungsten carbide removed by the laser (mm); *T* is the duration of laser processing (s).

The morphology of the grooves after 20 ablations and the geometrical characteristics are presented in Figure 4. As can be seen, the cross-section of the microslot texture could be divided into a U-shape and a V-shape, with a cross-sectional area that could be approximated by the area of a trapezoid or a triangle. The reason for this phenomenon is that laser energy conforms to a Gaussian distribution. As the laser processing depth gradually increases, the laser energy decreases, causing the laser processing area to gradually decrease. This results in a U-shaped cross-section in the microgroove. As the processing continues, the laser energy decreases again and ultimately results in a V-shaped cross-section with sharp and uneven contours instead of a flat bottom. The MRRs after different ablation cycles were calculated with the following equation, which was derived from Equation (1): (2)$$MRR=\frac{SL}{N}$$
where *N* is the number of ablation cycles, *L* (equal to the scanning speed) is the length of the ablation per unit time (mm/s), an *S* is the area of the cross-section of the groove after *N* ablations, (mm^2^). The areas and the average MRR per ablation are listed in Table 5. In Figure 5, it can be found that the increases in both laser power and laser frequency can effectively increase the MRR. In contrast, with the increase in the laser scanning speed, the MRR first increase sand then decreases. This indicates that a too-low or a too-high scanning speed reduces the MRR, leading to a decline in the laser processing efficiency. 

To further investigate the change in the MRR with ongoing laser ablation, the depth of the grooves after each ablation cycle was investigated. Figure 6 presents the change in the groove depth for different laser powers. It can be observed that the depth of the groove was larger when higher laser power was applied, and the depths obtained after 20 processing times were 12.99 µm (1.92 W), 21.15 µm (4.25 W), and 30.60 µm (7 W). Under the processing conditions of 4.25 W and 7.00 W, the groove depth linearly increased as the number of ablation cycles increased, while the MRR remained constant at an average value equivalent to that at N = 20 ablation cycles. However, when the power was 1.92 W, the increase in the depth for each ablation cycle gradually slowed down after 10 processing cycles, and the gradient was no longer a straight line; simultaneously, the MRR decreased. Considering the drastic increase in Ra (Figure 3), it was evident that at lower laser powers, the ablation processing became difficult, and the surface quality reduced with the continuation of the machining process. 

The influence of scanning speed was different from that of the laser power and laser frequency. Specifically (Figure 7), with the increase in the laser scanning speed, the depth after each ablation pass was smaller, and the depths were 15.76 µm (200 mm/s), 21.94 µm (150 mm/s), 30.60 µm (100 mm/s), and 68.25 µm (50 mm/s) after the experiments with 20 passes in each process. At the same energy density, changes in the laser scanning speed only affected the number of pulses within a certain ablation area. Therefore, it was obvious that the increase in the scanning speed reduced the number of pulses at the certain position, and this eventually led to the reduction in the amount of absorbed energy within this area. However, the reduction in laser energy per unit area did not result in a decrease in the MRR. As shown in Figure 5b, under the processing conditions with the highest scanning speed (50 mm/s), the MRR was even lower. Moreover, as the scanning speed increased, the change in the MRR did not monotonically increase. This indicated that it is necessary to select an optimal laser scanning speed within a certain range in order to maximize the MRR.

Similar trends in the change in the MRR were found when different laser frequencies were applied. As shown in Figure 8, with the increase in the laser frequency, the depth increased after 20 ablations, which were 15.35 µm (0.1 MHz), 30.60 µm (0.2 MHz), and 60.25 µm (0.5 MHz). The increase in laser frequency led to an increase in the equivalent number of pulses. When the single-pulse energy was constant, the laser influence also increased with increasing pulse frequency. However, the increasing trend in the processing depth became less linear after 15 (53.25 µm) and 20 (60.25 µm) ablations, with a pulse frequency of 0.5 MHz. This could be ascribed to the plasma shielding at the relatively high pulse frequency. When a laser beam is exerted on a workpiece surface, the facial material is vaporized due to the high-density energy, and the vapor continuously absorbs the laser energy, leading to the formation of high-temperature and high-density plasma on the target surface. The plasma rapidly expands outward and absorbs a proportion of the laser energy during the expansion process. Meanwhile, the plasma can act as a barrier that shielded the workpiece surface from laser radiation, thereby reducing the energy absorption of the workpiece materials. This phenomenon is known as the plasma shielding effect [38]. Additionally, the energy reduction caused by the absorption and scatters of the plasma, which is known as plasma refraction [39], can change the focusing shape of incident laser beams and reduce the utilization efficiency of laser energy. As shown in Figure 8, when the laser frequency and power were relatively high and the number of processing repetitions increased, the plasma shielding effect occurred on the surface of the material, resulting in a significant reduction in the MRR. Therefore, during laser processing processes, adequate adjustment of the laser power and frequency can avoid the occurrence of plasma shielding.

In summary, we confirmed that during the initial stage (N ≤ 10) of microgroove machining with different laser parameters, the laser energy was stable; due to the short processing time, the heat generated during processing could be dissipated without excessive accumulation. As a result, the MRR remained stable at a relatively high level. However, as the cycles continued (N > 10), lower laser power (1.92 W) could not continue the processing due to the lower laser energy. On the other hand, a slower laser scanning speed (50 mm/s) and a higher laser frequency (0.5 MHz) caused heat accumulation and plasma shielding, leading to a drop in the laser absorption rate and severe burning of the processing plane. This resulted in a significant decrease in processing depth as well a notable reductions in the surface quality and material removal rate. Therefore, selecting extreme processing parameter conditions is not an option to achieve a high MRR. This results in energy loss and a significant decrease in the energy efficiency ratio. In addition, from the above trends in the surface roughness and processing depth, it could be observed that a significant decrease in the processing depth occurs with a decrease in the surface roughness. Therefore, only by maintaining good surface morphology can processing continue to achieve the target conditions. 

### 3.2. Laser-Induced Periodic Surface Structure (LIPSS)

The microstructure of the materials can be examined by measuring the surface roughness and machined depth. The factors used to judge the quality of surfaces processed with a femtosecond laser can include the finer structures, such as the evolution of the nanostructures on the processed material’s surface. Therefore, in addition to the investigation of the microstructure, the nanostructures leading to the decline in MRR need to be investigated as well.

LIPSS is the surface nanostructure that can be produced via linearly polarized radiation on metals, semiconductors, ceramics, and polymers in ultrashort laser processing processes [40,41,42]. When solid materials are irradiated by polarized pulses near the material’s laser-induced damage threshold (LIDT), a series of periodic textures can be observed on the machined surface [43]. LIPSS includes low-spatial-frequency LIPSS (LSFL) for the wavelength order and high-spatial-frequency LIPSS (HSFL) for the subwavelength order [44,45]. In femtosecond laser machining, HSFL processing has a strong capability to produce high-efficiency and large-area uniform hierarchical nanotextures [46]. Typically, the direction of the LSFL texture is perpendicular to the polarization direction of the incident laser, whereas the direction the HSFL texture is parallel to the polarization direction of the incident laser [47]. 

In previous studies [48,49], the existence and integrity of the periodicity of LIPSS have been used as some criteria to assess the quality of the microstructure. As shown in Figure 9, some nanoscale structures could be observed on the surfaces of the grooves after laser ablation. When the laser frequencies were relatively low (0.1 MHz and 0.2 MHz), periodic nanowave patterns, i.e., LIPSS, were generated in the laser irradiation area (the red circles in Figure 9a,c). In contrast, when the laser frequency was 0.5 MHz, LIPSS could hardly be observed within the processed microgroove structure. Instead, due to the existence of uneven pits (Figure 9e,f), the Ra inside the groove significantly increased, demonstrating that a high-frequency laser was improper for the fabrication of the microstructure. 

Furthermore, LIPSS can influence the light absorption of a material and, eventually, the MRR. In some studies [50,51], the LIPSS increased the material’s absorption of lasers to over 90% or even close to 100%. This indicated that the MRR could be maintained at high level due to the existence of LIPSS because the nanostructure more effectively absorbs laser energy, which was proved by the evolution of the surface texture of Sample 8. Figure 10 shows the change in the LIPSS of Sample 8 during the 20 ablations. It can be seen that pillar-shaped LIPSS was generated on the workpiece surface after the first ablation, and the λ period was much smaller than the laser wavelength (1030 nm), which indicated that this structure was an HSFL. As the processing cycle continued, the LIPSS became more distinct, while its λ period remained unchanged. However, when the processing cycle was in the range of 8 ≤ N < 15, as mentioned earlier, the plasma shielding effect occurred, leading to a drop in the laser energy density on the material surface. At this stage, the LIPSS structure transitioned to LSFL, with its λ period being approximately equal to the laser wavelength. When the processing cycle N was ≥15, due to the accumulation of heat and other factors, the LIPSS completely disappeared, and the surface structure was destroyed. Taking into account the MRR (the curve of the 0.5 MHz line in Figure 8) and surface roughness (Figure 3h), it could be observed that the variation in the LIPSS strongly coincided with the other two curves. As the processing number increased, the LIPSS structure was destabilized, leading to an increase in the surface roughness within the microgroove and a decrease in the MRR. These findings indicated that the evolution of nanostructures is manifested in a more macroscopic form. Therefore, when complete LIPSS is obtained, the conditions for producing a high surface quality with a high MRR can be attained.

Similarly, Figure 11 shows the evolution of the LIPSS on the surface of Sample 3. It can be seen that the LIPSS was severely destroyed after 15 ablations, and the bottom of the groove exhibited uneven structures (Figure 11a). When the ablation test was finished, the LIPSS on the bottom surface had completely disappeared (Figure 11b). The corresponding effects on the MRR were reflected in the machining curve at the scanning speed of 50 mm/s (Figure 7a), where a reduction in the slope of the curve could be observed in the last 5 ablations (15 to 20), indicating the decrease in the laser processing rate. By comparing the two sets of laser processing conditions where the machining rate decreased, it could be seen that when the laser energy density was high and the scanning speed was slow (i.e., there was a large effective pulse number), repeated processing gradually generated heat accumulation, resulting in the destruction of the LIPSS, having a subsequent effect on the MRR. 

## 4. Conclusions

This study investigated the machining responses during the fabrication of microgroove textures on the surface of tungsten carbide using femtosecond laser ablation with various parameters. The evolutions of the surface roughness, geometry of the microtexture, and material removal rate with continuous laser machining were comprehensively investigated. The analysis revealed the dynamic relationship among the machining responses, processing efficiency, and laser-induced nanostructures on the surface of the WC materials. The findings provide a new perspective for investigating the surface quality and machining efficiency when fabricating microtextures on WC cutting tools with femtosecond lasers. The main conclusions reached in the research are as follows:

1. Increasing the laser power and laser frequency can effectively increase the MRR. However, the influence of laser scanning speed on the MRR does not follow a proportional increasing or decreasing relationship. Furthermore, there exists an upper speed limit, which falls between 100 and 150 mm/s. When the laser power and frequency are too high, plasma shielding effects may occur, causing a decrease in the energy reaching the workpiece surface and, subsequently, a decrease in the laser rate.

2. With the setup of suitable laser parameters (7.00 W and 0.2 MHz), an increase in processing duration can result in a more regular microgroove texture and a gradually more stable machining quality with lower roughness inside the groove. However, excessively small or large laser parameters can lead to a significant decrease in surface quality. The former is due to insufficient material removal caused by insufficient energy, while the latter is due to the destruction of the machining surface caused by plasma shielding, heat accumulation, and other factors that hinder the formation of the microgroove texture.

3. Clear LIPSS structures appear on the surface of tungsten carbide after the initial ablations, and LIPSS structures become increasingly apparent increases in the number of processing repetitions. Regular LIPSS enhances the light absorption of material surfaces, leading to a faster MRR. However, as the LIPSS is gradually destroyed, the MRR and surface roughness significantly decrease. Therefore, maintaining a good LIPSS structure is one of the conditions required to accelerate the MRR, improve efficiency, and achieve high-quality microgroove textures. 

## Figures and Tables

**Figure 1 micromachines-14-01143-f001:**
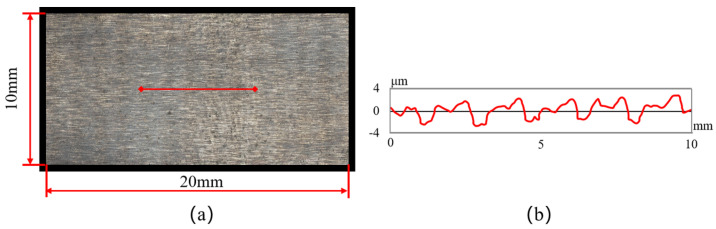
(**a**) Surface topography of tungsten carbide workpiece; (**b**) evolution of surface roughness.

**Figure 2 micromachines-14-01143-f002:**
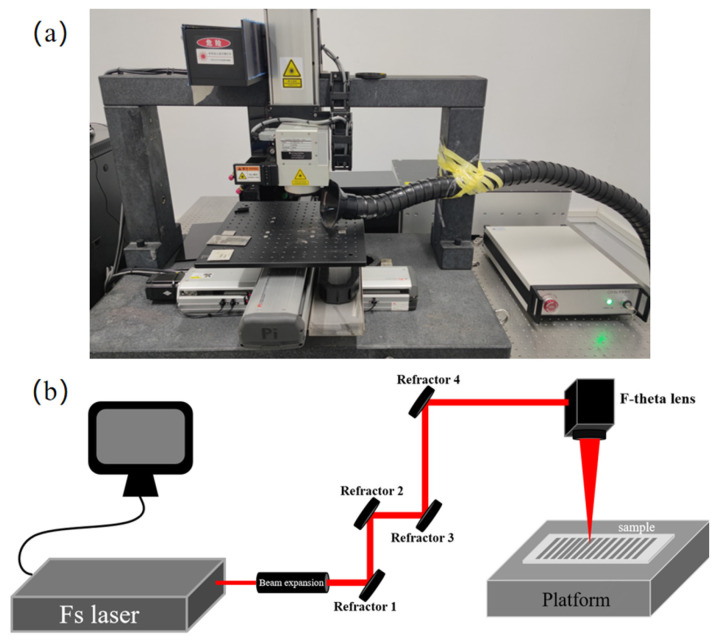
(**a**) Femtosecond laser experimental platform; (**b**) schematic diagram of femtosecond laser.

**Figure 3 micromachines-14-01143-f003:**
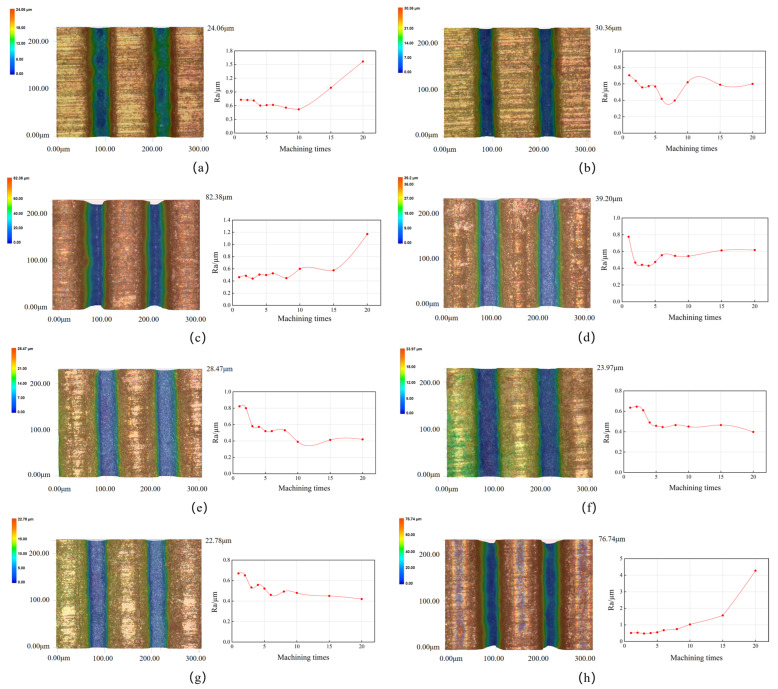
Morphology of tungsten carbide microgroove texture and variation in surface roughness of microgroove with increasing processing cycles: (**a**–**h**) Ras and morphology of Tests 1 to 8, respectively.

**Figure 4 micromachines-14-01143-f004:**
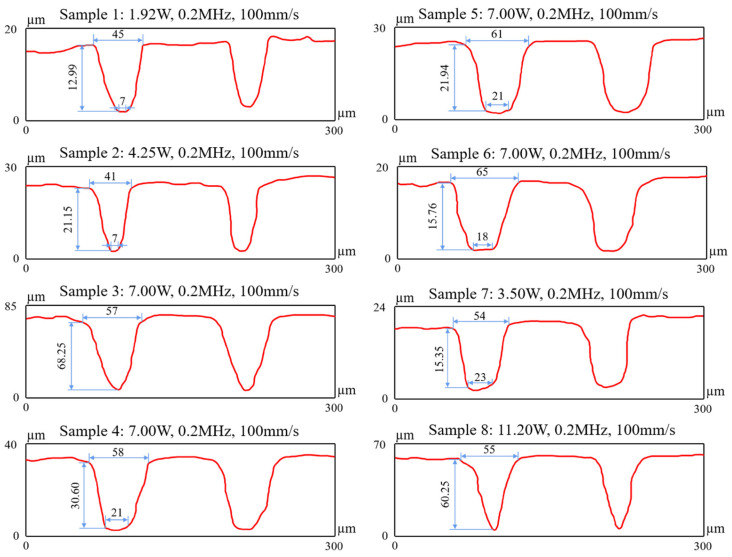
The morphology of grooves and geometrical characteristics after 20 processing cycles.

**Figure 5 micromachines-14-01143-f005:**
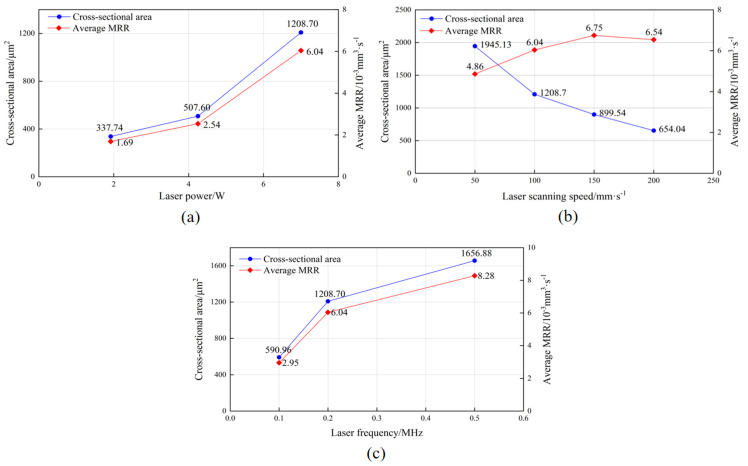
Variation trend in cross-sectional area and MRR of microslots with different parameters: (**a**) laser power: Samples 1, 2, and 4; (**b**) laser scanning speed: Samples 3, 4, 5, and 6; (**c**) laser frequency: Samples 4, 7, and 8.

**Figure 6 micromachines-14-01143-f006:**
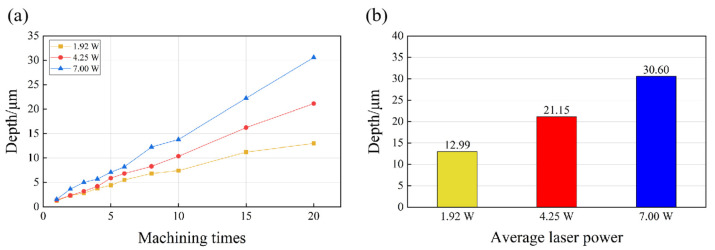
(**a**) The groove depth of the microgroove texture varied with the number of processing repetitions under different laser power conditions, and (**b**) the microgroove depth under different laser power conditions after 20 ablations.

**Figure 7 micromachines-14-01143-f007:**
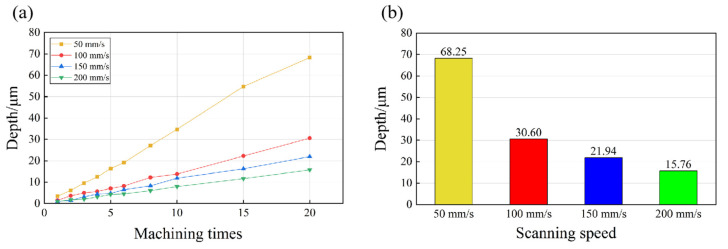
(**a**) The groove depth of the microgroove texture varied with the number of processing under different laser scanning speeds, and (**b**) the depth of the microgroove under different laser scanning speeds after 20 ablations.

**Figure 8 micromachines-14-01143-f008:**
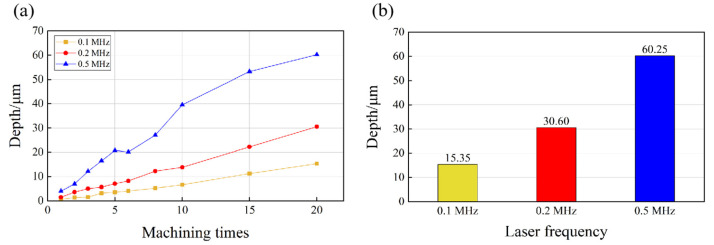
(**a**) Variation in groove depth of microgroove texture at different laser frequencies, and (**b**) microgroove depth at different laser frequencies after 20 ablations.

**Figure 9 micromachines-14-01143-f009:**
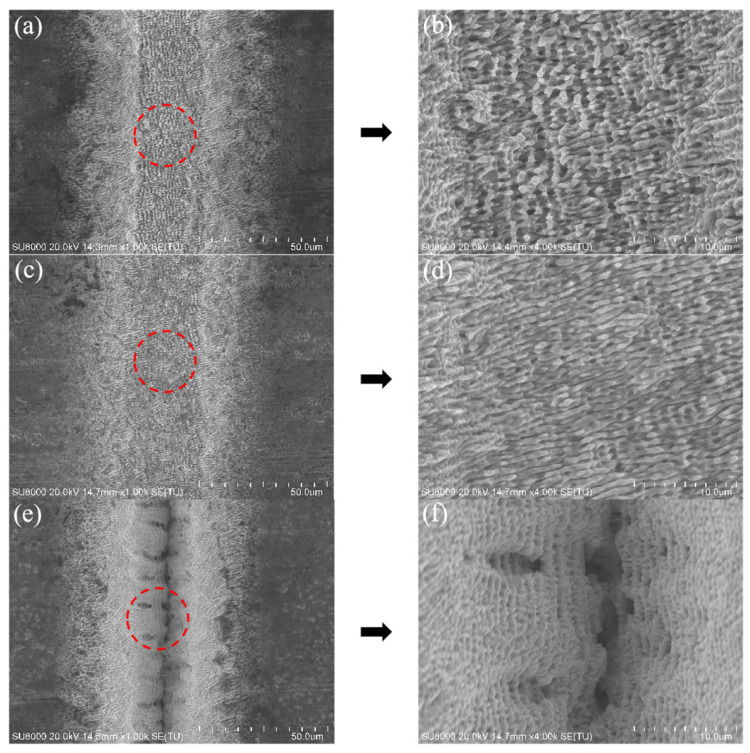
SEM images of the microgroove texture on the surface of tungsten carbide after 20 ablations at different laser frequencies: (**a**,**b**) 0.1 MHZ, (**c**,**d**) 0.2 MHz, and (**e**,**f**) 0.5 MHz.

**Figure 10 micromachines-14-01143-f010:**
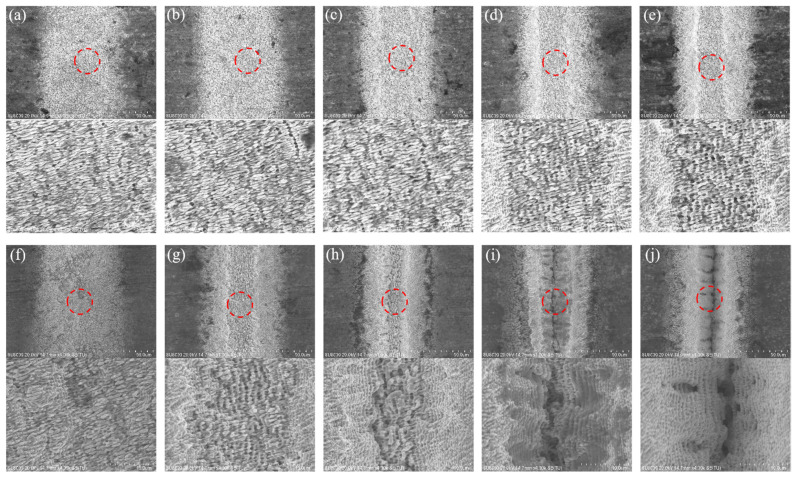
Surface LIPSS evolution of Sample 8 after 1–20 treatments: (**a**–**j**) LIPSS after the 1st to 6th, 8th, 10th, 15th, and 20th ablations, respectively.

**Figure 11 micromachines-14-01143-f011:**
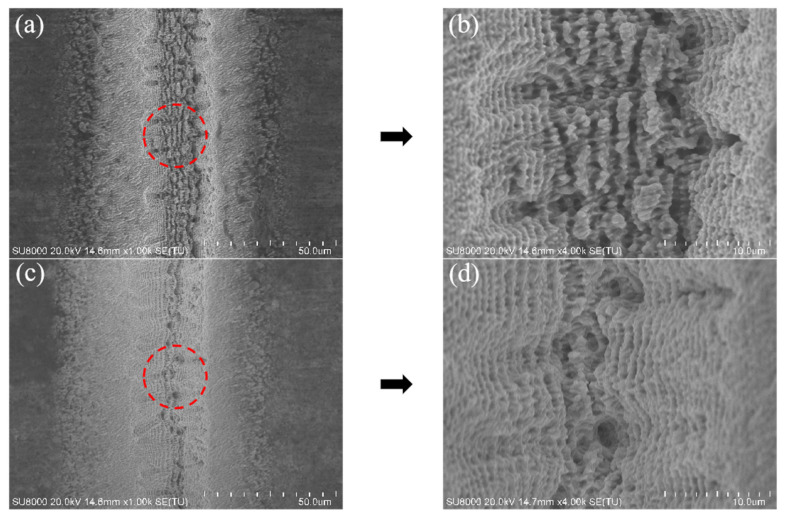
SEM images of the microgroove texture surface morphology of Sample 3 after the 15th (**a**,**b**) and 20th (**c,d**) ablations.

**Table 1 micromachines-14-01143-t001:** Mechanical properties of M20.

Parameter	Unit	Numerical Value
Density	g cm^−3^	12.4~13.5
Hardness	—	90.5
Bending strength	MPa	1350
Young’s modulus Thermal conductivity	GPa W/(m·K)	700 20~50

**Table 2 micromachines-14-01143-t002:** Chemical composition of M20.

Chemical Composition	Percentage (%)
WC	84
TiC + TaC	10
Co	6

**Table 3 micromachines-14-01143-t003:** The specifications of YSL PHOTONICS FemtoYL-20.

Parameter	Numerical Value
Central wave length (nm)	1030
Pulse width (fs)	300
Repetition frequency (kHz)	25~5000
Beam quality	M^2^ < 1.3

**Table 4 micromachines-14-01143-t004:** Laser parameters for processing tungsten carbide.

Sample No.	Average Laser Power (W)	Laser Frequency (MHz)	Scanning Speed (mm/s)	Ablation Cycles for the Measurement
1	1.92	0.2	100	1, 2, 3, 4, 5, 6, 8, 10, 15, 20
2	4.25	0.2	100	
3	7.00	0.2	50	
4	7.00	0.2	100	
5	7.00	0.2	150	
6	7.00	0.2	200	
7	3.50	0.1	100	
8	11.20	0.5	100	

**Table 5 micromachines-14-01143-t005:** The cross-section areas and the average MRR for microgroove machining.

Sample No.	Cross-Sectional Area (µm^2^)	Average MRR (10^−3^ mm^3^/s)
1	337.74	1.69
2	507.60	2.54
3	1945.13	4.86
4	1208.70	6.04
5	899.54	6.75
6	654.04	6.54
7	590.96	2.95
8	1656.88	8.28

## Data Availability

No new data were created or analyzed in this study. Data sharing is not applicable to this article.

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
