# Peer review of "Surface Quality and Material Removal Rate in Fabricating Microtexture on Tungsten Carbide via Femtosecond Laser"

_micromachines, 2023, doi:10.3390/mi14061143_

Round 1

Reviewer 1 Report

In the manuscript of micromachines-2402414, the authors fabricated a straight groove array micro-texture on the surface of WC tools via femtosecond laser with different machining parameters including laser power, laser frequency, scanning speed. And then MRR, surface roughness and the laser-induced periodic surface structure (LIPSS) were analyzed to reveal fundamental mechanisms enlighten the efficient machining method for the fabrication of micro-textures on ultrahard materials with ultrashort laser. The research is interesting, but the following comments are provided for the authors.

1. In section of introduction, the novelty of this research should be clarified deeply.

2. To check the problem of significant digit throughout the article.

3. The quality of Figure 3, 7, 8, 9 should be enhanced, in which the letters may be adjusted to be larger.

4. In the section of discussion,  the formation mechanism of laser-induced periodic surface structure (LIPSS) on WC should be discussed indetail.

Moderate editing of English language is required.

Reviewer 2 Report

The manuscript reported femtosecond laser surface micro-processing of tungsten carbide, and the influence of different machining parameters on the surface morphology of microgroove array. The material removal rate, surface roughness, and the LIPSS of tungsten carbide were studied and analyzed in details. Although a lot of experiments have been implemented, the experimental results and conclusions seem to be lack of creativity. I suggested that some experiments on investigating the relations between the surface structures and the cutting performance should be presented.

There are some issues that need to be addressed, as listed below.

1)    The data “density” listed in Table 1 is duplicated.

2)    Line 147, if there are references, please list them. Also, there is an extra space in this sentence.

3)    Please maintain consistency in the colors used for the two sets of data in Figure 4 (a) and (b). Line 238, the data “15.35 μm” is different from the data (16 μm) in Figure 6. Please confirm it.

4)    In the experimental section, line 311, why did Sample 7 not refer to its own test results (the curve of the 0.1 MHz line in Figure 6) but instead to the results of Sample 8 (the curve of the 0.5 MHz line in Figure 6)?

5)    Please refer to the recent progress on LIPSS induced by femtosecond laser processing (Adv. Mater. Interfaces 2023, 10, 2201924).

Some sentences in the text are quite complex, such as the last paragraph of the first part. It is recommended to use more concise expressions. Lines 352-354, the sentence does not read smoothly. 

Reviewer 3 Report

Dear Author(s), the manuscript ‘Surface quality and material removal rate in fabricating micro-texture on tungsten carbide via femtosecond laser’, Manuscript ID: micromachines-2402414, have some strong weakness that must be revised suitably.

Please find below some, of the most significant comments:

1.      Looking for the ‘Abstract’ section, it is well-written and interesting. However, I would suggest to omit presenting the shortcuts and abbreviations in this part.

2.      Similar to the previous content, the main objective of the studies proposed is not completely clear from the sentences presented. Please try to emphasize the required actions in resolving the mentioned issue, e.g. why material removal rate, roughness and the laser-induced periodic surface structure are considered? The common goal must be highlighted.

3.      The critical review in the ‘Introduction’ section is srongly limited. The Author(s) presented only  few brief lack in the last gap, lines 114-125 but, respectively, weakness of each of the previous studies should be even mentioned. From the current axis, the motivation looks more like proceeding in the previous studies not indicating some lacks in the current state of knowledge.

4.      In section 2, the Author(s) even mentioning about some previous experiments, e.g., ‘he laser frequencies and scanning speed were the optimal values obtained in our preliminary experiment and those published in literature.’, lines 146-147, selected some values like arbitrary. If any preliminary studies were arised, must be straightly referenced, if exist.

5.      Similar like in previous comment, there is no detail on the surface texture measurements. The SEM (scanning electron microscope), e.g., Hitachi UHR FE-SEM SU8000, is fraught with many errors, like most of the non-contact intruments. Please try to refer to the errors classified as outliers, noise, classify method of uncertainty reduction, like in:

(1)   https://www.doi.org/10.1088/2051-672X/3/3/035004

(2)   https://www.doi.org/10.3390/ma15155137

(3)   https://www.doi.org/10.1117/1.OE.59.6.064110

6.      In section 1.3, for some probes, selected parameters values, especially of processing/preparing the workpieces, were not justified. It must be presented in a more conscious way or suitably referenced to the previous analyses.

7.      Again for the section 3.1, equations (1) and (2) should be cited, if are not newly proposed by the Author(s). Even formulas are well-known and general, must be referenced to the primary sources, if exist.

8.      The subfigures in Table 5 must be classified according to selected order. Average MMR is not in order and, similarly, cross-section area (profile), valley depth, width. What was the criteria for presenting data in the received order? Currently, the meaning of the Table is lost.

9.      Generally, in section 3.1, the Author(s) did not present any critical sentences. In fact, the discussion does not occur and, respectively, only the advantages are included but the disadvantages must be defined as well.

10.  In section 3.2 it is diffult to follow was was new received. Mixing the current results with those received and published previously makes the reader confused. Please try to emphasize the new studies separately, or even highlight those encouraging the current prospects.

11.  The ‘Conclusion’ section is good, present proper presentation of the main purposes. However, the Author(s) should indicate one, the main proposals for highlighting the nowelty used. In its current form, it is difficult to separate this issue.

Generally, the proposed manuscript has some weaknesses that must be minimised so, respectively, in the current form is not suitable for publication in a quality journal as the Micromachines is.

The manuscript must be improved significantly before any further processing, if allowed by the Editor.

Round 2

Reviewer 1 Report

Dear authors, thank you for all the responses that were addressed properly and make the manuscript suitably improved for publication in Micromachines as it is.

Reviewer 2 Report

Publish as it is

Reviewer 3 Report

Dear Author(s), the manuscript titled ‘Surface quality and material removal rate in fabricating micro-texture on tungsten carbide via femtosecond laser’, Manuscript ID: micromachines-2402414, has been improved suitably so, respectively, can be further processed by the Micromachines journal.

Thank you for all of your full responses that, in their current form, were addressed properly and make the manuscript suitably improved for publication in a quality journal as the Micromachines is.